# PREGEN: UNCOVERING LATENT THOUGHTS IN COMPOSED VIDEO RETRIEVAL

## ABSTRACT

Composed Video Retrieval (CoVR) aims to retrieve a video based on a query video and a modifying text. Current CoVR methods fail to fully exploit modern Vision-Language Models (VLMs), either using outdated architectures or requiring computationally expensive fine-tuning and slow caption generation. We introduce *PREGEN* (*PRE GENeration extraction*), an efficient and powerful CoVR framework that overcomes these limitations. Our approach uniquely pairs a frozen, pre-trained VLM with a lightweight encoding model, eliminating the need for any VLM fine-tuning. We feed the query video and modifying text into the VLM and extract the hidden state of the final token from each layer. A simple encoder is then trained on these pooled representations, creating a semantically rich and compact embedding for retrieval. *PREGEN* significantly advances the state of the art, surpassing all prior methods on standard CoVR benchmarks with substantial gains in Recall@1 of +27.23 and +69.59. Our method demonstrates robustness across different VLM backbones and exhibits strong zero-shot generalization to more complex textual modifications, highlighting its effectiveness and semantic capabilities.

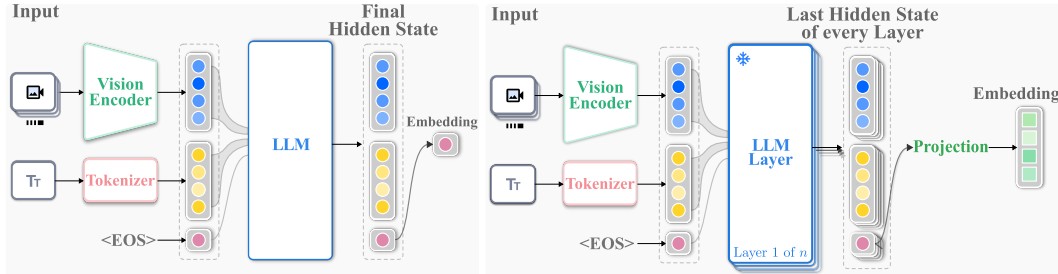

(a) Other VLM-based embedding methods        (b) *PREGEN* (Ours)

Figure 1: Comparison of other CoVR approaches, which finetune the VLM and use only the last layer embedding, and *PREGEN*, which freezes the VLM and aggregates embeddings from all layers.

## 1 INTRODUCTION

Video understanding has been a fundamental challenge in computer vision, with early work focusing on action recognition (Wang et al., 2011; Wang & Schmid, 2013), video classification (Karpathy et al., 2014; Yue-Hei Ng et al., 2015), and object detection (Girshick et al., 2014; Redmon et al., 2016). As video content continues to grow across digital platforms, the need for effective video retrieval has become increasingly important. However, traditional keyword-based and visual similarity approaches face challenges in effectively capturing complex user intents. Composed Video Retrieval (CoVR) (Ventura et al., 2024a) addresses this challenge by defining the task of retrieving a target video given both a reference video and a modifying text query, where a query expresses a specific modification to a reference video, such as "find a video similar to this cooking demo, but with a different cuisine."

CoVR builds upon Composed Image Retrieval (CIR), which established the foundational approach of using both visual and textual inputs for retrieval tasks. Early work like TIRG (Vo et al., 2018) developed image-text fusion architectures for CIR. Fashion-IQ (Wu et al., 2020) introduced the first CIR dataset with human-written relative captions, and demonstrated that combining reference images with natural language modifications enables more effective image retrieval. Subsequent work expanded CIR to general domains with datasets like CIRR (Liu et al., 2021). Recent zero-shot approaches like Pic2Word (Saito et al., 2023) and LinCIR (Gu et al., 2024) have leveraged vision-language models such as CLIP (Radford et al., 2021) for composed retrieval without task-specific training.

However, the transition from images to videos introduces substantial complexity that existing CIR methods fail to address (Nguyen et al., 2024). Unlike static images, videos contain temporal sequences with dynamic scene changes, object interactions, and narrative progressions that demand processing significantly richer semantic content (Simonyan & Zisserman, 2014; Goyal et al., 2017). These challenges require novel architectures that can efficiently capture both the temporal dynamics of video content and the nuanced modifications expressed in natural language (Nguyen et al., 2024; Xu et al., 2021).

Modern Vision-Language Models (VLMs) like LLaVA (Liu et al., 2023) and Qwen-VL (Bai et al., 2023) have demonstrated remarkable capabilities in understanding complex visual scenes and their relationships to natural language descriptions. VLMs are trained on vast amounts of image-text and video-text pairs, and possess rich world knowledge about objects, scenes, actions, and their interactions. These strengths make VLMs a natural foundation for CoVR, which requires understanding how textual modifications relate to video content.

While significant progress has been made in curating large-scale datasets for CoVR, including WebVid-CoVR (Ventura et al., 2024a), FineCVR (Yue et al., 2025), and Dense WebVid-CoVR (Thawakar et al., 2025), the field faces a significant methodological gap. Current approaches fail to fully exploit the rich world knowledge encoded in modern VLMs. Existing methods either use outdated models based on CLIP-style (Radford et al., 2021) joint-encoders (Yue et al., 2025; Hummel et al., 2024), require extensive fine-tuning of large models (Kong et al., 2025), or rely on computationally expensive caption generation processes that create bottlenecks for practical deployment (Thawakar et al., 2024; Hummel et al., 2024). This autoregressive caption generation approach becomes particularly costly when scaling to large video databases with millions of videos.

To address these limitations, we introduce *PREGEN* (*PRE GENeration extraction*), a novel framework that efficiently leverages frozen VLMs for CoVR without requiring fine-tuning or caption generation. Our key insight is that hidden states from the final token across all VLM layers contain complementary semantic information that, when properly aggregated, creates powerful embeddings for video retrieval. We extract these multi-layer representations and process them through a lightweight Transformer encoder to produce compact yet semantically rich embeddings. Figure 1 illustrates our approach compared to other VLM-based methods.

**Our contribution.**

- We observe that existing VLM-based embedding methods rely on single-layer representations. This approach results in embeddings that fail to capture the full knowledge encoded across the VLM.

- Motivated by this observation, we propose *PREGEN*, an efficient embedding framework that extracts hidden states from the final token across all VLM layers and aggregates them using a lightweight Transformer encoder. This approach makes better use of VLM knowledge while avoiding expensive fine-tuning or caption generation.

- We introduce a training strategy that precomputes hard negative batches by grouping queries whose reference videos come from the same source. This approach preserves the benefits of hard negatives while eliminating the costly process of online similarity search.

- We achieve significant state-of-the-art improvements with substantial performance gains. We conduct an extensive ablation study to empirically validate the sources of our strong results, demonstrating the effectiveness of our multi-layer pooling approach.

## 2 RELATED WORK

**Composed Image Retrieval (CIR).** Composed Image Retrieval (CIR) has emerged as a fundamental task in multimodal information retrieval. In CIR, the goal is to search for target images using a composition of a reference image and a text modifier that describes desired changes. Wu et al. (2020) and ComposeAE (Anwaar et al., 2021) explored image-text fusion architectures and established early methods for learning joint image-text embeddings for CIR. More recent methods, such as InstructCIR (Zhong et al., 2024) and MagicLens (Zhang et al., 2024a), have utilized VLMs to handle more complex and nuanced compositional modifications and achieve zero-shot performance without task-specific training.

**Composed Video Retrieval (CoVR).** The task of CoVR was introduced by Ventura et al. (2024a), who released the first CoVR dataset, WebVid-CoVR, and established initial baselines using naive frame sampling paired with CLIP-based encoders. Notably, this simple approach struggles at capturing important temporal information, due to the static nature of CLIP-like architectures. Yue et al. (2025) used temporal pooling mechanisms and demonstrated improvements in temporal understanding. Notably, they also intoduced FineCVR, a more complex CoVR dataset where modifications require a deeper understanding of video dynamics. This improves on WebVid-CoVR, where queries can often be resolved using a single frame. Hummel et al. (2024) proposed a caption-based method that generates intermediate text descriptions before matching. However, this approach creates computational bottlenecks of slow autoregressive text generation. UNITE (Kong et al., 2025) fine-tuned VLMs for multimodal retrieval tasks, using LoRA (Hu et al., 2022) to achieve competitive results without full retraining. Still, most current approaches either rely on outdated CLIP architectures or require expensive computational resources, leaving significant room for more efficient methods that fully leverage modern VLM capabilities.

**Vision Language Models (VLMs).** VLMs have revolutionized multimodal understanding by learning unified representations across visual and textual modalities. Early approaches like CLIP (Radford et al., 2021) and BLIP (Li et al., 2022) introduced dual-encoder and encoder-decoder architectures for cross-modal alignment and understanding. Modern VLM work has increasingly moved toward generative architectures that demonstrate superior reasoning capabilities. Modern generative VLMs such as the LLaVA series (Liu et al., 2023; Zhang et al., 2025) and Qwen-VL series (Bai et al., 2023; Wang et al., 2024) have shown strong performance in complex multimodal tasks. By leveraging LLM backbones and sophisticated vision-language alignment techniques, VLMs are capable of using the vast real-world knowledge of LLMs to perform complex visual reasoning, generate detailed descriptions, and understand intricate visual-textual relationships.

In the context of CoVR and CIR, these generative models offer advantages in understanding nuanced compositional queries. Their autoregressive nature allows for more flexible reasoning about visual-textual relationships, making them particularly well-suited for tasks that require compositional understanding. Methods like (Thawakar et al., 2024) and TFR-CVR (Hummel et al., 2024) use VLMs to generate captions for the query and target video, thus crossing the modality gap of vision and text. Other methods like InstructCIR (Zhong et al., 2024) and UNITE (Kong et al., 2025) directly use the hidden representations of the VLM as embeddings for retrieval.

**Universal multimodal retrievers and LMM-based retrieval.** Recent work has explored universal retrievers that can handle text, images, and mixed inputs within a single embedding space. These methods convert multimodal LLMs into bi-encoders that generate unified embeddings for multiple data modalities. E5-V (Jiang et al., 2024) extends the original E5 (Wang et al., 2022) framework to multimodal settings by aligning different input modalities in a shared embedding space. The method converts visual inputs into structured text descriptions and trains on instruction-formatted data pairs to learn cross-modal alignments. MM-Embed (Lin et al., 2025) builds on the NV-Embed (Lee et al., 2025) architecture, using modality-aware hard negative mining, and fine-tuning to improve performance on text retrieval benchmarks. GME (Zhang et al., 2024b) trains a multimodal embedder on both real and synthetically generated image-text pairs, studying how model size and training data volume affect retrieval accuracy across different tasks.

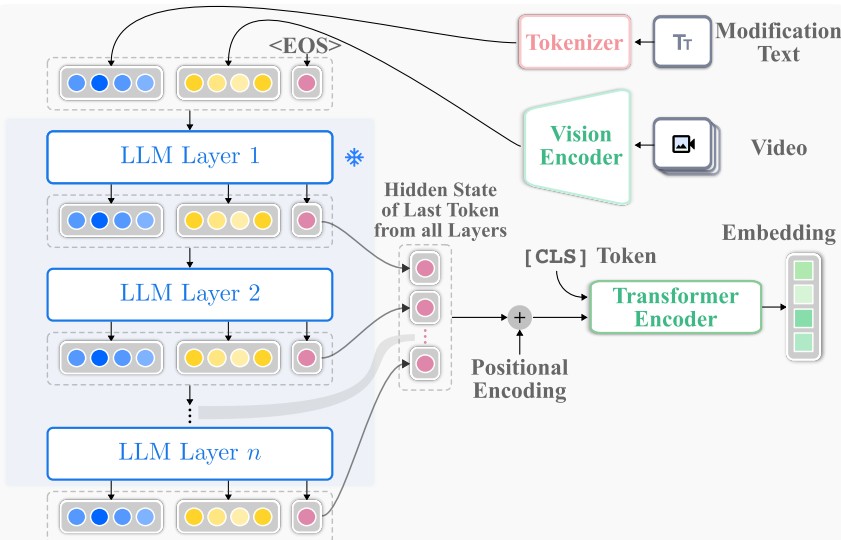

Figure 2: *PREGEN* extracts the hidden state of the last token at every VLM layer. These vectors are position encoded and fed into a Transformer Encoder with a [CLS] token. The [CLS] token output is then projected through an MLP to produce the final embedding.

## 3 METHOD

### 3.1 PROBLEM

Composed Video Retrieval (CoVR) is the task of retrieving a target video from a large database given a query consisting of a reference video and a modifying text description. Unlike standard video retrieval, which matches queries to targets based on visual similarity, CoVR requires understanding how the query video should be modified based on the text. Formally, let $\mathcal{V}_q$ be a database of query videos, and $\mathcal{V}_t$ a database of target videos. Given a query composed of a reference video $v_q \in \mathcal{V}_q$ and modifying text $t_m$ that describes desired changes to the reference video, the goal is to retrieve the target video $v_t \in \mathcal{V}_t$ that best satisfies the modification specified by $t_m$ when applied to $v_q$.

The training data consists of triplets $(v_q, t_m, v_t)$ where $v_q$ is the query video from $\mathcal{V}_q$, $t_m$ is the natural language modification text, and $v_t$ is the corresponding target video from $\mathcal{V}_t$ that matches the described changes. The modifying text $t_m$ typically describes transformations such as changes in objects ("replace the guitar with a piano"), scenes ("move from outdoors to indoors"), actions ("walking instead of running"), or other semantic modifications that preserve the core context of the reference video. During inference, given a new query $(v_q, t_m)$, the model ranks all videos in $\mathcal{V}_t$ by their similarity to the composed query.

### 3.2 MODEL ARCHITECTURE

Current CoVR methods that leverage VLMs typically extract embeddings from only the final layer, missing the hierarchical knowledge encoded across different layers of the model. Recent works show that different layers in a VLM capture distinct semantic information. Early layers capture low-level visual and textual features, middle layers learn compositional relationships between visual and textual signals, and encode global information, and later layers integrate information for high-level reasoning and next token prediction, thus depending far less on raw visual tokens (Tao et al., 2024; Liu et al., 2024; Chen et al., 2024). To fully utilize this multi-layer knowledge, we propose *PREGEN*, which extracts and aggregates representations from all VLM layers. The full architecture is illustrated in figure 2.

Given a query video $v_q$ and modifying text $t_m$, we first process them through a frozen VLM to obtain joint video-text representations. Let $f_{VLM}$ denote the VLM with $L$ layers. We feed the

concatenated input $[v_q; t_m]$ through the VLM and extract the hidden state of the final token from each layer $l \in \{1, 2, \ldots, L\}$, yielding $\boldsymbol{h}_l \in \mathbb{R}^d$ where $d$ is the hidden dimension of the VLM.

The multi-layer features

$$\boldsymbol{H} = [\boldsymbol{h}_1, \boldsymbol{h}_2, \ldots, \boldsymbol{h}_L]$$

are first concatenated to a learnable $\boldsymbol{h}_{cls}$ token. We then apply sinusoidal positional encodings based on layer order, where earlier layers receive lower positional indices and later layers receive higher indices. Specifically, we compute $\tilde{\boldsymbol{h}}_l = \boldsymbol{h}_l + PE(l)$ for each layer representation, and $\tilde{\boldsymbol{h}}_{cls} = \boldsymbol{h}_{cls} + PE(0)$ for the cls token, forming the input sequence

$$\tilde{\boldsymbol{H}} = [\tilde{\boldsymbol{h}}_{cls}, \tilde{\boldsymbol{h}}_1, \ldots, \tilde{\boldsymbol{h}}_L].$$

This encoding scheme enables the Transformer encoder to better distinguish between representations from different VLM layers and understand their sequential ordering.

The Transformer encoder takes as input the sequence $\tilde{\boldsymbol{H}}$ and produces the attended representations

$$\boldsymbol{Z} = f_{VLM}(\tilde{\boldsymbol{H}}) = [\boldsymbol{z}_{cls}, \boldsymbol{z}_1, \cdots, \boldsymbol{z}_L].$$

We extract the output of the encoder for the $\tilde{\boldsymbol{h}}_{cls}$ token, which serves as an aggregated representation of all layer information. Finally, we project $\boldsymbol{z}_{cls}$ through an MLP to obtain the final embedding:

$$\boldsymbol{e} = MLP(\boldsymbol{z}_{cls}),$$

where $\boldsymbol{e} \in \mathbb{R}^D$ is the $D$-dimensional embedding used for retrieval.

Target video embeddings are generated using the same process, with the target video $v_t$ as the sole input to the VLM. During retrieval, we compute cosine similarity between the query embedding $\boldsymbol{e}_q$ and each target video embedding $\boldsymbol{e}_t$ in the database to rank candidates.

## 3.3 TRAINING

We train *PREGEN* using contrastive learning. The goal is to learn an embedding space where semantically similar query-target pairs have high similarity while dissimilar pairs have low similarity. We also introduce a novel hard negative mining approach that leverages the inherent structure of CoVR datasets, allowing for efficient hard negative mining without the usual burden of additional computational overhead.

**Contrastive learning.** Given a batch of of triplets $\mathcal{B} = \{(v_q^i, t_m^i, v_t^i)\}_{i=1}^B$, the goal is to maximize the similarity of each query embedding $e_q^i$ to its corresponding target embedding $e_t^i$, while minimizing the similarity between the query embedding and all other target embeddings in the batch. Pairs of the form $(e_q^i, e_t^i)$ are called positive pairs, while pairs of the form $(e_q^i, e_t^j)$ such that $i \neq j$ are called negative pairs.

We use the symmetric InfoNCE loss (Oord et al., 2018):

$$\mathcal{L} = \frac{1}{2B} \sum_{i=1}^B \left[ -\log \frac{\exp(s_{ii}/\tau)}{\sum_{j=1}^B \exp(s_{ij}/\tau)} - \log \frac{\exp(s_{ii}/\tau)}{\sum_{j=1}^B \exp(s_{ji}/\tau)} \right].$$

where $s_{ij}$ denotes the cosine similarity of $e_q^i$ and $e_t^j$, and $\tau$ is a temperature hyperparameter. InfoNCE is widely used for retrieval tasks, as it aims to maximize the similarity between positive pairs while simultaneously minimizing the similarity between all negative pairs in the batch.

**Hard negative mining using source-based batching.** Hard negative mining is a training technique that selects challenging negative examples for training. Specifically, instead of using random negative samples, hard negative mining identifies negatives that are difficult for the current model to distinguish from positive examples; typically, those with high similarity to the positive samples. By training on these difficult cases, the model learns a more discriminative embedding space, ultimately resulting in more robust representations that achieve better performance on retrieval tasks. However,

hard negative mining typically requires expensive online computation to identify challenging examples during training. We propose a preprocessing approach that achieves similar benefits without any additional computational overhead. During data preprocessing, we prioritize grouping training triplets that share the same query video $v_q$ when constructing batches. This ensures that batches contain multiple triplets with the same source video but different modifications whenever possible. Since target videos from the same source share visual similarity, they naturally serve as challenging negatives that improve the model's discriminative capability without requiring explicit hard negative search. For an empirical validation of the benefits of this approach, see Section 4.2.

# 4 EXPERIMENTS

In this section, we conduct experiments aiming to answer these core questions:

- **Q1** Does aggregating representations from all layers improve retrieval performance compared to using a single layer?
- **Q2** Does source-based hard negative mining improve model capabilities?
- **Q3** Is the method agnostic to different backbones?
- **Q4** Does the method generalize to more complex and detailed textual modifications without additional training?
- **Q5** Do the different components of *PREGEN* contribute to its performance?

All experiments are conducted on 4 NVIDIA RTX PRO 6000 Blackwell Workstation Edition, using an AdamW optimizer. Full hyperparameter details, and dataset statistics are provided in Sections A and B. All results for *PREGEN* are reported using a Qwen2.5-VL 7B backbone (Bai et al., 2025) unless stated otherwise.

Our codebase is publicly available at: `https://anonymous.4open.science/r/PREGEN_CoVR`.

## 4.1 THE EFFECT OF USING ALL LAYERS

**Setup.** To evaluate the impact of using all layers (**Q1**), we compare *PREGEN*, as described in Section 3.2, to *PREGEN* when trained using only the last layer of the VLM. In this case, since there is no longer a sequence of hidden states, we discard the Transformer encoder, and directly process the hidden state through an MLP. We perform evaluation on two available CoVR datasets: WebVid-CoVR (Ventura et al., 2024a), and FineCVR (Yue et al., 2025), reporting Recall@$k$ for $k \in \{1, 5, 10, 50\}$. For WebVid-CoVR we report the baselines CoVR (Ventura et al., 2024b), CoVR-2 Ventura et al. (2024a), ECDE (Thawakar et al., 2024), Thawakar et al. (2025), and UNITE (Kong et al., 2025), taken from (Kong et al., 2025). For FineCVR we report the baselines CoVR (Ventura et al., 2024b), TFR-CVR (Hummel et al., 2024), FreestyleRet (Li et al., 2024), and FDCA (Yue et al., 2025), taken from (Yue et al., 2025).

**Results.** Tables 1 and 2 establish *PREGEN* as state-of-the-art on WebVid-CoVR and FineCVR, surpassing all other CoVR methods by huge margins. Notably, *PREGEN* outperforms the previous state-of-the-art, UNITE (Kong et al., 2025), a method which also uses the hidden states of VLMs to generate embeddings. Conversely, when using a single layer of the VLM, *PREGEN* performs poorly, achieving the weakest results across all metrics for both datasets. This large disparity in performance is a direct result of utilizing the full scope of information encoded across the layers of the VLM, further validating the strength of our approach.

## 4.2 THE EFFECT OF SOURCE-BASED HARD NEGATIVE MINING

**Setup.** To evaluate the impact of our source-based hard negative mining strategy (**Q2**), we compare *PREGEN* trained with and without hard negative mining. We evaluate both configurations on WebVid-CoVR (Ventura et al., 2024a) and FineCVR (Yue et al., 2025), reporting Recall@$k$ for $k \in \{1, 5, 10\}$.

Table 1: Results on WebVid-CoVR test set. The best results are highlighted.

| Method | R@1 | R@5 | R@10 | R@50 |
|---|---|---|---|---|
| CoVR | 53.1 | 79.9 | 86.9 | 97.7 |
| CoVR-2 | 59.8 | 83.8 | 91.3 | 98.2 |
| ECDE | 60.1 | 84.3 | 91.3 | 98.7 |
| UNITE$_{instruct}$ 7B | 72.5 | 90.8 | 95.3 | 99.5 |
| *PREGEN* (1 layer) | 14.20 | 30.71 | 40.18 | 67.45 |
| *PREGEN* | **99.73** | **99.92** | **99.96** | **100.00** |

Table 2: Results on FineCVR test set. The best results are highlighted.

| Method | R@1 | R@5 | R@10 | R@50 |
|---|---|---|---|---|
| TFR-CVR | 15.21 | 40.12 | 52.78 | 81.75 |
| CoVR | 17.05 | 41.57 | 56.60 | 85.56 |
| FreestyleRet | 20.39 | 52.98 | 68.37 | 93.03 |
| FDCA-CLIP | 25.84 | 55.84 | 70.23 | 94.33 |
| FDCA-BLIP | 26.79 | 63.21 | 78.65 | 97.25 |
| *PREGEN* (1 layer) | 6.90 | 12.21 | 17.42 | 39.95 |
| *PREGEN* | **96.38** | **99.95** | **99.98** | **99.99** |

**Results.** Table 3 demonstrates the gains added by using our source-based hard negative mining strategy. On WebVid-CoVR, hard negative mining consistently improves performance across all metrics, with gains of 0.98%, 1.17%, and 1.21% for Recall@$\{1,5,10\}$ respectively. The improvements are substantially more pronounced on FineCVR, where the method provides performance boosts of 16.05%, 2.98%, and 1.22% across the same metrics. This could suggest that our hard negative mining approach may prove even more effective on more challenging datasets where performance has not reached saturation levels. We emphasize that this training strategy incurs no additional computational costs, and can potentially be used across many more domains and tasks.

Table 3: Results of *PREGEN* with and without source-based hard negative mining. Best results are in bold.

| Method | WebVid-CoVR | | | FineCVR | | |
|---|---|---|---|---|---|---|
| | R@1 | R@5 | R@10 | R@1 | R@5 | R@10 |
| *PREGEN* (No hard negative mining) | 98.75 | 98.75 | 98.75 | 80.33 | 96.97 | 98.76 |
| *PREGEN* | **99.73** | **99.92** | **99.96** | **96.38** | **99.95** | **99.98** |

### 4.3 THE EFFECT OF DIFFERENT BACKBONES

**Setup.** To evaluate whether our method is agnostic to different VLM backbones (**Q3**), we compare *PREGEN* using four different Qwen-VL variants: Qwen2-VL 2B, Qwen2-VL 7B, Qwen2.5-VL 3B, and Qwen2.5-VL 7B (Wang et al., 2024; Bai et al., 2025). All models follow the same architecture described in Section 3.2, with only the underlying VLM backbone being changed. We evaluate all variants on WebVid-CoVR (Ventura et al., 2024a), reporting Recall@$k$ for $k \in \{1, 5, 10, 50\}$. We report baseline results for UNITE (Kong et al., 2025) using Qwen2-VL 2B and Qwen2-VL 7B backbones, taken from (Kong et al., 2025).

**Results.** Table 4 demonstrates that *PREGEN* achieves consistently strong performance across all tested VLM backbones, with minimal variation in results despite the differences in model size and

Table 4: Results on WebVid-CoVR using different VLM backbones. The *Backbone* column specifies the underlying VLM used for each method. Best results are in bold.

| Method | Backbone | R@1 | R@5 | R@10 | R@50 |
|---|---|---|---|---|---|
| UNITE$_{instruct}$ | Qwen2-VL 2B | 69.1 | 88.4 | 93.2 | 99.1 |
| UNITE$_{instruct}$ | Qwen2-VL 7B | 72.5 | 90.8 | 95.3 | 99.5 |
| *PREGEN* | Qwen2-VL 2B | 95.89 | 98.28 | 98.59 | 99.53 |
| *PREGEN* | Qwen2-VL 7B | 98.51 | 98.71 | 98.71 | 98.75 |
| *PREGEN* | Qwen2.5-VL 3B | 97.81 | 99.61 | 99.8 | 99.96 |
| ***PREGEN*** | Qwen2.5-VL 7B | **99.73** | **99.92** | **99.96** | **100.00** |

Table 5: Results on Dense WebVid-CoVR test set. *Train Data* column indicates the dataset used for training. Best results are in bold.

| Method | Train Data | R@1 | R@5 | R@10 | R@50 |
|---|---|---|---|---|---|
| Thawakar et al. (2025) | WebVid-CoVR | 48.08 | 73.36 | 81.06 | 93.78 |
| Thawakar et al. (2025) | Dense WebVid-CoVR | 71.26 | 89.12 | 94.56 | **98.88** |
| ***PREGEN* (Ours)** | WebVid-CoVR | **98.71** | **98.75** | **98.79** | 98.79 |

architecture. All *PREGEN* variants substantially outperform existing state-of-the-art methods, with performance differences between backbones remaining within $4\%$ across all metrics. Notably, comparing performance between the 2B and 7B variants of Qwen2-VL, our method shows an average relative decrease of only $1.1\%$ across Recall@$k$ for $k \in \{1, 5, 10\}$, while UNITE shows a $3.2\%$ average relative decrease. This consistency across diverse backbone sizes and versions further demonstrates the robustness of our approach, regardless of the underlying VLM architecture.

### 4.4 GENERALIZATION TO COMPLEX INSTRUCTIONS

**Setup.** Thawakar et al. (2025) published the Dense WebVid-CoVR dataset, a modified version of the WebVid-CoVR dataset, where textual modifications are more fine grained. Unlike the original dataset's simple textual modifications (e.g., "change color to red"), Dense WebVid-CoVR contains detailed compositional descriptions that specify precise spatial relationships, temporal dynamics, and multi-object interactions within video scenes. To test generalization to complex instructions (**Q4**), we train *PREGEN* exclusively on the standard WebVid-CoVR. We then evaluate on the Dense WebVid-CoVR test set to assess its ability to handle significantly more detailed modifications. We use the baselines of Thawakar et al. (2025), reporting Recall@$k$ for $k \in \{1, 5, 10, 50\}$.

**Results.** Table 5 reveals that *PREGEN* demonstrates exceptional generalization to more complex textual instructions, experiencing less than $2\%$ performance decrease across all metrics when moving from standard to dense modifications. In contrast, Thawakar et al. (2025) suffer a substantial performance decrease, with drops as high as $23\%$ compared to their results on the original WebVid-CoVR dataset. This demonstrates the utility of our approach in handling diverse and complex instructions, without requiring retraining. We attribute this robustness to our use of a large VLM backbone, which provides significantly greater flexibility in understanding nuanced textual descriptions compared to the simpler BLIP backbone used by Thawakar et al. (2025).

### 4.5 EFFECT OF DIFFERENT COMPONENTS

**Setup.** To evaluate how much each component contributes to the strong performance of the method (**Q5**), we conduct an ablation study, testing variants of PREGEN with: (1) single-layer extraction instead of multi-layer, (2) averaging encoder outputs instead of using the [CLS] token, (3) removal

Table 6: Ablation study results on WebVid CoVR. The best results are highlighted.

| Method | R@1 | R@5 | R@10 | R@50 |
|--------|-----|-----|------|------|
| *PREGEN* (1 layer) | 14.20 | 30.71 | 40.18 | 67.45 |
| *PREGEN* (avg over encoder outputs) | 98.71 | 98.75 | 98.79 | 98.79 |
| *PREGEN* (No hard negative mining) | 98.75 | 98.75 | 98.75 | 98.75 |
| *PREGEN* (No positional encodings ) | 98.75 | 98.75 | 98.79 | 98.79 |
| *PREGEN* | 99.73 | 99.92 | 99.96 | 100.00 |

of hard negative mining, and (4) removal of positional encodings. All experiments are conducted on the WebVid-CoVR test set.

**Results.**    Table 6 demonstrates the importance of each component in *PREGEN*. As shown in our main results, using only a single layer leads to severe performance degradation. Averaging encoder outputs instead of using the [CLS] token, training without hard negative mining, and removing positional encoding, all lead to a small but consistent decrease in performance, validating our different design choices.

## 5    DISCUSSION

While *PREGEN* achieves nearly perfect performance on current CoVR benchmarks, these results should not be interpreted as indicating that composed video retrieval is a solved problem. Rather, we believe these results indicate that current available benchmarks for CoVR do not mirror the challenges posed by real-world video understanding tasks. For example, the WebVid-CoVR dataset often contains modifications that can be resolved using a single frame, rather than requiring understanding of temporal dynamics or complex semantic relationships Thawakar et al. (2024). FineCVR attempts to address this limitation by including more complex modifications. Indeed, performance on this dataset was substantially lower compared to WebVid-CoVR (96.38% vs 99.73% Recall@1). This difference highlights that future benchmarks should prioritize modifications that require more advanced compositional reasoning, temporal causality, and complex spatial-temporal relationships.

Our work reveals that current embedding and retrieval methods underutilize modern VLMs. Most approaches use outdated architectures or extract features from single layers, thereby neglecting meaningful semantic information. We believe future research should explore how to better utilize the different layers of VLM and LLM backbones, investigate semantic information distribution across model layers, and develop more efficient aggregation strategies. These findings can and should be adopted across other multimodal tasks where foundation models are underutilized. Similar multi-layer extraction strategies could improve capabilities in tasks such as image-text retrieval, visual question answering, and cross-modal understanding.

## 6    CONCLUSION

We introduced *PREGEN*, a novel framework for Composed Video Retrieval that efficiently leverages frozen vision-language models by extracting and aggregating hidden states from all layers of the VLM. In contrast to previous methods that used VLMs, our approach eliminates the need for expensive fine-tuning or caption generation while achieving substantial performance improvements. Namely, *PREGEN* surpasses all prior methods on the standard CoVR benchmarks WebVid-CoVR and FineCVR, with gains of +27.23 and +69.59 in Recall@1, respectively.

We introduce a training strategy we term source-based hard negative mining, which utilizes the structure of CoVR datasets and improves training efficiency without additional computational overhead. *PREGEN* exhibits strong performance across different VLM backbones and generalization to complex textual modifications, highlighting the robustness of our multi-layer feature aggregation.

## REPRODUCIBILITY STATEMENT.

We have taken care to ensure the reproducibility of our results. Complete details of datasets, model architectures, training settings, and hyperparameters are provided in the main text and Appendix. All datasets used are publicly available benchmarks. All pretrained VLMs are taken from Hugging-Face repositories. We provide a public codebase with a complete implementation of all methods presented.

## ETHICS STATEMENTS.

This work does not involve human subjects, personal or sensitive data, or applications in high-risk domains. All datasets used in our experiments are publicly available benchmarks, and were used in compliance with their respective licenses. The large-language models used in our study are publicly released open-source models obtained through HuggingFace. Our work provides a general framework for Composed Video Retrieval, and does not target any harmful applications.

**Usage of Large Language Models in This Work.** LLMs were used in this work for coding assistance, grammar refinement, and LaTeX formatting. Their use allowed the authors to invest more time into meaningful research contributions.

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

# A  HYPERPARAMETERS

Table 7: Training and model hyperparameters used for each dataset.

| | WebVid-CoVR | FineCVR |
|---|---|---|
| Learning rate | 0.00005 | 0.00005 |
| Weight decay | 0.05 | 0.05 |
| Batch size | 1024 | 1024 |
| Epochs | 1 | 1 |
| Gradient clipping norm | 1.0 | 1.0 |
| Dropout | 0.1 | 0.1 |
| Number of frames (uniformly sampled) | 8 | 8 |
| Encoder heads | 8 | 8 |
| Transformer encoder number of layers | 1 | 1 |
| MLP number of layers | 2 | 2 |
| MLP hidden dimension | 14,336 | 14,336 |
| InfoNCE temperature $\tau$ | 0.05 | 0.05 |
| Precision | bfloat16 | bfloat16 |

# B  DATASET STATISTICS

Table 8: Dataset statistics.

| | WebVid-CoVR | FineCVR |
|---|---|---|
| Train triplets | 1,648,789 | 1,000,028 |
| Test triplets | 2,556 | 10,043 |
| Videos | 133,219 | 136,547 |
| Unique words | 21,098 | 20,961 |

