# OpenReview forum: "PREGEN: Uncovering Latent Thoughts in Composed Video Retrieval"
_ICLR.cc/2026/Conference — ICLR 2026 Conference Withdrawn Submission_

### Official Review · Reviewer_7LY5 · 2025-10-29

**Soundness:** 3
**Presentation:** 2
**Contribution:** 2
**Rating:** 4
**Confidence:** 4

**Summary:**

The authors study the problem of composed video retrieval (CoVR), claiming that the existing CoVR methods are limited in many ways: outdated architectures, computational limitations, slow caption generation. PREGEN is proposed to address the current limitations, where fine-tuning, caption generation is not required. More specifically, PREGEN relies on pre-trained VLM with lightweight encoding, both of which are kept frozen. One of the key points is fully utilizing each layer, where previous work has been using the final token in the last layer only. When training the model, hard negative mining strategies have been implemented for further improvements. The experimental results highlight the efficacy of PREGEN.

**Strengths:**

- The problem formulation is clear. The limitations in existing studies have been well discussed, and PREGEN is proposed to address those limitations.
- Experiments are well designed to support the claims. The authors show how using every layer boost the performances. Hard negative mining also brings additional improvements which has been explicitly shown in Table 3.
- PREGEN achieves near 100% already in R@1.

**Weaknesses:**

- The main idea of using every VLM layer is somewhat simple, which can be easily thought of/ or easily tried as the first attempt. Specifically, PREGEN with avg. over encoder outputs already achieves significant improvements. While this is a nice finding, it also shows that the proposed scheme does not require technical efforts (less technical challenge). It is also surprising that 1 layer PREGEN performs poorly compared to previous approaches with 1 layer, which raises many questions.
- While the authors utilizes VLM, the scope of the study is limited only focusing on CoVR. When compared to UNITE which is the VLM-based model, the contribution of PREGEN is limited. Besides, UNITE is not specifically designed for CoVR, yet it performs impressively utilizing only the final layer. The authors can provide further discussions on this.
- The authors never mentioned the computational limitations that might be cased by the full-ayer computations in VLM, which is layer connected to Transformer Encoder. Does this still beat previous models with *slow caption generation* methods in speed ?

**Questions:**

Q1. How efficient is PREGEN (full) when compared to PREGEN (1 layer) ?

Q2. Overall, is there any trade-off between the performance and computational complexity from using the full layers in PREGEN?

Q3. Is there any reason why PREGEN underperforms when using only 1 layer?

---

### Official Review · Reviewer_5zJ8 · 2025-10-31

**Soundness:** 3
**Presentation:** 3
**Contribution:** 2
**Rating:** 4
**Confidence:** 4

**Summary:**

This paper focus on the task of composed video retrieval. PREGEN extracts and aggregates hidden states from all layers of a frozen vision-language model (VLM) to produce semantically rich video–text embeddings. Experiments on WebVid-CoVR, FineCVR, and Dense WebVid-CoVR show that PREGEN achieves strong performance.

**Strengths:**

1, The motivation behind their design is well explained.
2, Demonstrates strong performance.
3, Solution is easy and makes sense to work.

**Weaknesses:**

1, The multi-layer pooling to bring rich feature is intuitive, thus brings less novelty.
2, Overclaim of no-training or finetuning. l85 is misleading.The aggregated feature is then projected by training.
3, The generalisation to ood setup is not clear. Section 4.4 carry experiment on WebVid dataset. seems not a OOD scenario.
4, Written is not smooth, for example,  the latent thoughts in the title is not defined or discussed.

**Questions:**

1, How does PREGEN behave on unseen domains, is the light transformer encoder is able to handle this situation?
2, Could aggregating hidden states helps as well when finetuing the LLM like what other methods do?
3, What does the latent thoughts mean.

---

### Official Review · Reviewer_U79f · 2025-11-03

**Soundness:** 2
**Presentation:** 3
**Contribution:** 2
**Rating:** 4
**Confidence:** 4

**Summary:**

This paper proposes PREGEN, an efficient composed video retrieval framework that leverages multi-layer representations from a frozen Vision-Language Model (VLM).

Instead of using only the final-layer features, PREGEN aggregates hidden states across multiple layers to form a semantically richer embedding.

The approach avoids VLM fine-tuning and introduces a lightweight encoder for retrieval. Experimental results demonstrate **remarkable improvements**, with Recall@1 reaching up from 72% to 98% on standard CoVR benchmarks.

**Strengths:**

1. The paper is well-written and clearly presented, making it easy to follow the motivation and design choices.

2. The **retrieval performance is impressive**, significantly outperforming prior methods on several CoVR benchmarks.  Improve previous results from  26.79 to 96.38 on FineCVR.

3. Using frozen VLMs with a lightweight encoder is an efficient design philosophy, aligning with current trends in leveraging pre-trained multimodal models. The work provides some evidence of robustness across backbones and textual modifications.

**Weaknesses:**

The Recall@1 = 98% result **seems unusually high**. There is no analysis on potential overfitting or data leakage, which undermines the credibility of the claim. __Can you check again if the evaluation code is correct?__ For example, test a subset from MSRVTT/MSVD.

The methodological novelty is limited: the core idea mainly extends existing MLLM-based retrieval pipelines by aggregating multi-layer features rather than relying solely on the last layer.

The paper lacks an efficiency analysis — inference and computational cost remain unclear, especially since modern MLLMs are heavy for retrieval tasks.

The comparison only contrasts single-layer vs. multi-layer aggregation; there is no systematic exploration (e.g., curve showing performance vs. number of layers used). The paper could benefit from more diagnostic experiments (e.g., cross-domain tests or noise robustness) to justify the claimed generalization ability.

**Questions:**

Could the authors provide a computation cost comparison (e.g., FLOPs, inference latency) between PREGEN and other retrieval baselines?

How does performance evolve as more layers are aggregated? A trend curve (number of layers vs. Recall@1) would clarify the contribution of multi-layer fusion.

Have the authors verified data splits to rule out potential leakage or overlap between training and test sets, given the unusually high Recall@1 results?

Can PREGEN generalize to retrieval tasks with domain shift (e.g., from activity datasets to instructional videos)? Have you consider the common used benchmarks like MSRVTT/MSVD?

Would incorporating intermediate fusion strategies (e.g., attention-weighted layer aggregation) further improve the representation efficiency?

I **may reconsider my voting if all my concern are addressed during the rebuttal section**.

---

### Official Review · Reviewer_it1Y · 2025-11-03

**Soundness:** 3
**Presentation:** 2
**Contribution:** 1
**Rating:** 2
**Confidence:** 4

**Summary:**

This paper proposes an efficient and powerful framework for Composed Video Retrieval.

**Strengths:**

The paper introduces an efficient and well-designed framework for composed video retrieval.

**Weaknesses:**

1. **Writing issues** – The long sentence in line 95 (“This approach results in embeddings that fail…”) is confusing and should be rewritten for clarity.
2. **Figures** – Figures 1 and 2 lack proper legends. For instance, the meanings of the blue, yellow, and pink circles are unclear.
3. **Architecture confusion** – In line 231, Figure 2 shows that hidden states are encoded by a *transformer encoder*, but the text mentions another *VLM* used for processing. This inconsistency should be clarified.
4. **Limited novelty** – The method mainly adopts a standard contrastive loss, which is commonly used in retrieval-related works. The novelty of the training objective appears limited.
5. **Missing inference details** – The paper does not clearly explain how inference is conducted. During training, video and text embeddings are fused, but the target dataset only provides video embeddings. Please clarify how retrieval works at test time.
6. **Dataset and metrics** – The paper lacks a clear and separate description of the datasets used and the evaluation metrics applied.

**Questions:**

Please refer to the points listed under *Weaknesses*.

---

### Note · Authors · 2025-12-02

I have read and agree with the venue's withdrawal policy on behalf of myself and my co-authors.